



# Brief communication: RADIX (Rapid Access Drilling and Ice eXtraction) dust logger test in the EastGRIP hole

Authors: Jakob Schwander[1,2], Thomas F. Stocker[1,2], Remo Walther[1,2], Samuel Marending[1,2], Tobias Erhardt[1,2,3], Chantal Zeppenfeld[1,2], Jürg Jost[4]

[1]Climate and Environmental Physics, Physics Institute, University of Bern, Sidlerstrasse 5, 3012 Bern, Switzerland
[2]Oeschger Centre for Climate Change Research, University of Bern, Hochschulstrasse 4, 3012 Bern, Switzerland
[3]now at Institute of Geosciences and Frankfurt Isotope and Element Research Center (FIERCE), Goethe University Frankfurt, Altenhöferallee 1, 60438 Frankfurt am Main, Germany
[4]Spacetek Technology AG, Brüggliweg 18, 3073 Gümligen, Switzerland

*Correspondence to*: Thomas F. Stocker (thomas.stocker@unibe.ch)

Abstract: The RADIX (Rapid Access Drilling and Ice eXtraction) optical dust logger is part of the exploratory drilling system developed at the University of Bern. It was previously untested because no RADIX borehole reached to the depth of the required bubble-free ice. In June 2023, we tested the logger with an adapter for the large EastGRIP (East Greenland Ice-core Project) deep borehole. An excellent dust record was obtained for the Bølling-Allerød-Younger Dryas-Early Holocene period. The light scattered by the dust in the ice around the borehole was slightly higher than the detection range of the logger, requiring a reduction in the sensitivity for future deployments.

## 1 Introduction

The RADIX (Rapid Access Drilling and Ice eXtraction) system designed and developed at the University of Bern consists of
a 40 mm diameter shallow drill for the installation of casing and packer in the firn, a hydraulically driven 20 mm diameter deep drill and a logger for inclination, azimuth, temperature and dust in the ice surrounding the borehole. For details of the system, refer to the publications by Schwander et al. (2014; 2023). While logging inclination, azimuth and temperature were successfully tested in the RADIX holes at Little Dome C in Antarctica, dust logging was not possible because so far we had never reached the depth beyond the transition from bubbles to clathrate. The optical dust logger is based on a design by Bay
et al. (2001). It emits light by a 405 nm LED and measures the backscattered light from the dust particles in the ice by a photomultiplier tube (PMT). It is designed to operate in bubble free ice. In bubbly ice the reflected light is in a range that saturates the amplifier.

## 2 Methods






**Figure 1: Top: RADIX logger and 3.5mm fiber-optic cable winch: A marks the emitting 405 nm LED; B marks the receiving window and mirror (figure from Schwander et al. (2023)). Bottom: logger mounted in the adapter. The parts are colored for better visibility. The actual adapter is deep black anodized or painted, with the exception of the spring wires. Focus cones of the LED and the receiving mirror are shown in the right panel. Simulations showed that most of the detected photons are scattered only once.**
**The intersection of the cones thus determines the vertical resolution of the dust logger, which is of the order of 0.2 m.**



To test the dust logger without the need to drill a new hole with the RADIX drill, we constructed an adapter for testing the logger in an existing deep hole of larger diameter (Fig. 1). The main purpose of the adapter is to hold the logger close to the borehole wall with the emitting LED and the receiving optics in the rectangular direction to the wall, and to minimize straylight inside the hole. The RADIX logger is mounted in a tube connecting two aluminum blocks, each with a spring wire on the back that presses the block against the opposite borehole wall, whereby the hole and adapter radii are precisely matched. The backscattering of light from dust has been simulated numerically at CSEM (Centre Suisse d'Electronique et de Microtechnique) with monosized particles of 2 μm diameter (Schwander et al., 2023). The simulated mean path length of the scattered light from a light source to a planar detector placed in the same plane is of the order of 0.5 m. Most of the captured light is only scattered once. Based on the simulation we designed the optics of the logger as shown in Fig. 1. The LED points downward at an angle of 30° (22° when taking refraction into account). The receiving mirror focuses 40° upwards.

As the dust concentration in the ice varies over two orders of magnitude between the warm, less dusty and the cold dusty periods, the logger needs to be able to cover a wide range of backscattered light. To cover the expected range, the output power of the LED, as well as the high voltage of the PMT, are cyclically switched between two values. While the LED power is switched every 0.1 s, the PMT voltage changes only every 5 s because it needs a longer settling time. In this setup, a total of four sensitivity ranges are alternatively available within 10 s. The total sensitivity range is approximately –49 to –111 dB. The sensitivity ranges were determined based on the specifications of the LED and the PMT and a geometrical transmission of 0.017, given by the ratio of solid angles of the receiving PMT area and the LED radiation cone. These ranges are as follows: range 1: –49 to −76 dB; range 2: –59 to −86 dB; range 3: –74 to −101 dB; range 4: –84 to −111 dB. A comparison between the sensitivity of the logger and the expected scattered light intensity was shown in Schwander et al. (2023), but not all geometric details were considered there. To compare the sensitivity with the simulations, where no geometric transmission was taken into account, we add $10$ dB$*\log(0.017) \approx -18$ dB to the above ranges, resulting in a total range of approximately –67 to –129 dB. The simulated signal is −60 to −83 dB for the Antarctic Plateau. It is based on a distance of 0.2 m between the LED and PMT with a 10 mm diameter receiving area. Correcting for the actual dimensions of the logger (0.38 m LED-PMT distance and 8 mm diameter receiving area) results in an approximately 5 dB lower signal, i.e. −65 to −88 dB for the Antarctic Plateau, slightly exceeding the calculated lower sensitivity limit of the dust logger. Since the logger was primarily designed for use in Antarctica, where dust levels are about an order of magnitude lower than in Greenland, we expected that the high glacial dust levels in Greenland would possibly exceed the measuring range of the dust logger. Although the sensitivity ranges could be changed, we refrained from doing so, because it was the first test of the logger in bubble free ice, and the expected levels of light scattered by the dust was so far only model based and is subject to expected uncertainties, e.g. due to the monosized particles as compared to the natural distribution.



# 3 Field Work and Results

In June 2023, during the EastGRIP project, we could carry out a logging run in the deep borehole, drilled to a depth of 2640 m at that time. The East Greenland Ice-core Project includes retrieving an ice core by drilling through the Northeast Greenland Ice Stream with the aim to gain new knowledge on ice stream dynamics and past climate. We deployed the logger

using the RADIX winch with 3 km of fiber-optic cable fed through the sheaves with depth encoder of the EastGRIP drill tower.

We started descending at 100 mm/s. Due to friction of the adapter on the wall and in the fluid we had to reduce the descending velocity gradually to avoid slack in the cable. Between 1000 and 1425 m the velocity was about 40 mm/s. Down to 1100 m depth the dust signal was in saturation due to the strong backscatter by the bubbles. Between 1100 and 1200 m a

gradual drop of the backscattered light indicated the transformation of the air bubbles to clathrates. At 1425m depth we reached full glacial ice (age approx. 16 ka) and due to the high dust level we observed permanent saturation of the dust signal and decided to stop and pulled up with a hoisting speed of about 70 mm/s.

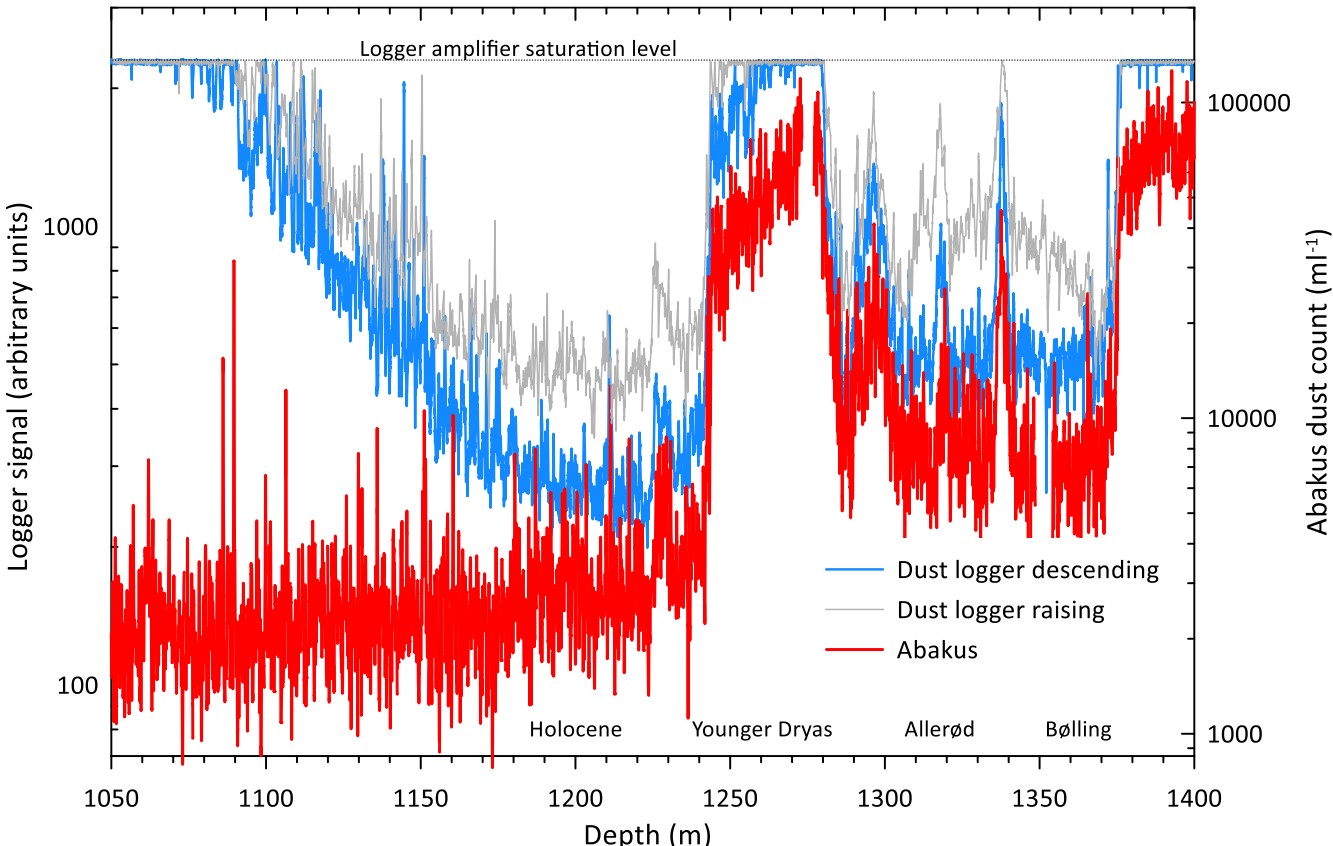

**Figure 2: Optical dust logger record compared with the Abakus particle detector data at 0.15 m resolution.**



From the transition into Bølling (depth ca. 1375 m) to the early Holocene (depth ca. 1170 m), except for the first half of Younger Dryas, where the signal was in saturation, the obtained record is of good quality (Fig. 2). We compare the logger signal with the number of particles measured on the ice core by an Abakus particle detector (range 1 to 15 μm), which is part of the continuous flow analysis (CFA) system in Bern (Erhardt et al., 2023; Kaufmann et al., 2008). Since the fiber-optic cable is only 3.5 mm in diameter it is rather stretchable. Also, we observed some slip on the encoding sheave when

descending. Accordingly, the depth reading was off by few meters. The depth scale was therefore linearly adjusted to match the transition into Bølling (GIS 1e) and the end of the Younger Dryas (GS 1).

## 4 Discussion and Conclusions

We notice an excellent correspondence between the signal of the RADIX optical dust logger and the CFA particle count, especially in case of the record during descent. While hoisting the signal level was somewhat higher and more irregular. We

suspect that either ice cuttings that are floating in the fluid accumulated on the adapter or the adapter was slightly pulled away from the wall, or a combination of both. The Abakus data have been smoothed by a Gaussian filter (0.15 m full with at half maximum). At this resolution the variance of the dust count data within climatically stable depth intervals matches the variance of the logger data, in agreement with the estimated depth resolution of the RADIX dust logger (Fig. 1). Fig. 3 depicts a detailed logger record between 1335.5 and 1337 m. The record shifts smoothly between climatically caused

changes in the dust level, which is a consequence of the averaging by the optical geometry (Fig. 1), but also shows that instrumental noise is low.



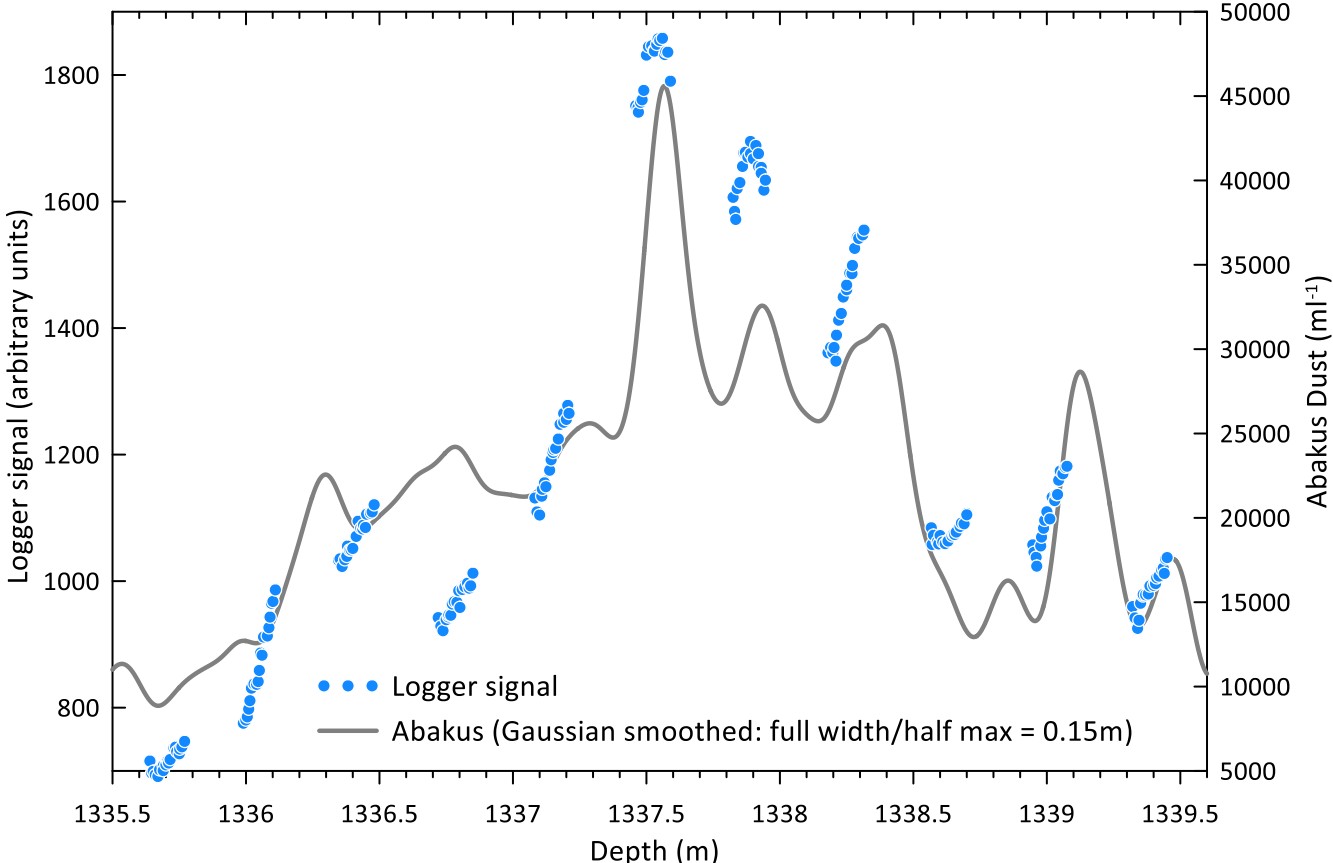

**Figure 3: Details of the dust logger record of a prominent dust peak (Older Dryas). Gaps without data occur during the intervals with high voltage at the photomultiplier (high sensitivity range, amplifier in saturation).**

The dust signal plotted in Fig. 2 was measured in the least sensitive range of the logger. The ranges of higher sensitivity were all close to or in saturation. That is, even with the lower dust concentrations in Antarctica, the signal would be within the two least sensitive ranges. The highest dust values in the Antarctic Plateau (Delmonte et al., 2002) roughly correspond to the values of the Younger Dryas in the interior of Greenland (Simonsen et al., 2019), where we observe partial saturation. This is in very good agreement with the simulated range, slightly exceeding the sensitivity range of the logger. A less sensitive

range is therefore also required for the Antarctic, and in order to cover all dust levels in Greenland the measuring range needs to be extended by about 15 dB on the low sensitivity side.

Despite the fact that the fluid in the EastGRIP hole was quite milky, as observed by a borehole camera (D. Dahl-Jensen pers. comm.), the adapter seemed to block stray light sufficiently. However, we cannot completely rule out the possibility that stray light in Antarctic boreholes would be a problem at the 10-fold lower dust concentrations during the warm periods.

However, when using the logger in the narrow 20-mm RADIX hole flushed with filtered fluid, stray light is expected to be significantly lower. After the current Beyond EPICA deep drilling at Little Dome C, there is an opportunity to use the



logger, again with an adapter for the larger borehole diameter, in Antarctica and obtain a dust record over many glacial cycles (Lilien et al., 2021).

*Acknowledgements.* We would like to thank Hanspeter Moret for implementing the controller software. We are grateful to Jörg Pierer, Centre Suisse d'Electronique et de Microtechnique, for providing an extensive simulation of the light scattering on dust particles in ice. We are indebted for the field support at the EastGRIP project offering the opportunity for testing the logger in the deep bore hole. EastGRIP is directed and organized by the Centre for Ice and Climate at the Niels Bohr Institute, University of Copenhagen.

*Financial support.* This research has been supported by the Swiss National Science Foundation (SNSF) (project nos. 159563, 172745 and 200492), Frederik Paulsen of Ferring International and the Fondation Prince Albert II de Monaco (project no. 2436). EastGRIP is supported by funding agencies and institutions in Denmark (A. P. Møller Foundation, University of Copenhagen), USA (US National Science Foundation, Office of Polar Programs), Germany (Alfred Wegener Institute, Helmholtz Centre for Polar and Marine Research), Japan (National Institute of Polar Research and Arctic Challenge for Sustainability), Norway (University of Bergen and Trond Mohn Foundation), Switzerland (Swiss National Science Foundation), France (French Polar Institute Paul-Emile Victor, Institute for Geosciences and Environmental research), Canada (University of Manitoba) and China (Chinese Academy of Sciences and Beijing Normal University).

*Author contributions.* JS developed and designed the overall concept of RADIX. RW and SM improved design details and realized mechanical components. JS, TS and RM participated in field campaigns in Greenland and Antarctica and provided photographs. JJ led the manufacturing of the logger. TE and CZ measured and prepared the CFA dust data. JS wrote the manuscript with contributions from all co-authors.

*Competing interests.* The contact author has declared that none of the authors has any competing interests.

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
