# Peer review of "Brief communication: RADIX (Rapid Access Drilling and Ice eXtraction) dust logger test in the EastGRIP hole"

_EGUsphere, 2024_

## Author Response (AR1)

**Authors responses to Reviewer1 Comments on egusphere-2024-372**

Reviewer 1: Ryan Bay
* * *
"Brief communication: RADIX (Rapid Access Drilling and Ice eXtraction) dust logger test in the EastGRIP hole" by Jakob Schwander et al., The Cryosphere Discuss.
* * *
**Schwander et al. give a note showing some tantalizing results using a dust logger in the EastGRIP, Greenland borehole. The RADIX dust logger principle and design are partly based on our tools, and I'm delighted to see these results.**

**The RADIX logger was intended for use in small RADIX boreholes.  For this work the tool was fitted with adapters for use in the much larger EastGRIP borehole. Part of innovation of the RADIX drill system is the very narrow 2 cm borehole, resulting in greater mobility and reduced drilling fluid. This compact design places fairly severe space constraints on borehole instruments.**

**The RADIX design uses a light source which is less focused and canted downward, departing from a tight focus directed sidewise. The PMT detector has a limited acceptance angle, and the authors deem the depth resolution of the tool to be ~20 cm based on the intersection of the source-receiver focus cones.  Judging from Figure 3 the resolution may be better than this.  This geometry also potentially impacts the dynamic range of the logger over greatly varying polar dust concentrations, and Schwander et al. have considered this including use of simulations.  In the cleanest Antarctic ice, the effective scattering length can be tens of meters from AMANDA/IceCube, so this geometry might be less effective there. In a narrow RADIX borehole, fluid drag and logging speed might also pose issues.**

Reply: Due to the special geometry of the overlapping focus cones (Fig. 1), the depth resolution is expected to be largely independent of the scattering length. We do not expect the measurement speed to influence the resolution, apart from a certain depth shift due to the elongation of the cable.

**A reader is left wanting to see more data than the four meters of logger-core comparison shown in Figure 3, perhaps over a greater depth range.**

We agree in principle, but the number of figures is limited to 3 in a brief communication. We show the entire record in Fig. 2, and in the detailed figure 3 we have selected a part of the data set with a strongly varying signal to demonstrate the response sensitivity of our logger, both for signal increase and signal decrease.

**At Summit, Greenland in the GISP2/GRIP boreholes, we found that bubbles had not completely converted to clathrates and disappeared until below 1600 m.  So I wonder if the ice depths at EastGRIP where these logger measurements were made are 100% bubble-free.  Dust logging in bubbly ice is achievable (ref. 1), but more dependent on conditions and less easy to interpret than clear ice.**

The good agreement of the logger signal in Fig. 2 with the measured dust values below 1200m indicates that the influence of scattered light from bubbles is practically negligible here, which corresponds very well to the disappearance of the bubbles in the EastGRIP ice core as documented by visual inspection using line scanning (Weikusat, I., Westhoff, J., Kipfstuhl, S., and Jansen, D.: Visual stratigraphy of the EastGRIP ice core (14 m–2021 m depth, drilling period 2017–2019), PANGAEA [data set], https://doi.org/10.1594/PANGAEA.925014, 2020.). This paper is now cited.

Since the main focus of the RADIX logger is to obtain a fast qualitative dust record of the deep part of an ice sheet for the purpose of age estimation and quality control of the stratification for site prospection, the acquisition of dust data of the bubbly ice and the absolute dust content are not expected from RADIX in the current version.

**Authors responses to Reviewer2 Comments on egusphere-2024-372**

Reviewer 2: Matthias Hüther
* * *
"Brief communication: RADIX (Rapid Access Drilling and Ice eXtraction) dust logger test in the EastGRIP hole" by Jakob Schwander et al., The Cryosphere Discuss.
* * *
**General comments:**

**Schwander et al. presents first data from a new dust logger designed for narrow boreholes. The test was conducted in a larger borehole as designed, forcing the use of an adapter, and in a different location (Greenland) with higher dust concentration compared to the intended operating area in Antarctica. The report focuses on the evaluation of the optical dust measurement of the scattering from dust particles, as the scattering from bubbles saturates the sensitivity range. Therefore, it does not allow the measurement of dust absorption from dust in non-bubble free ice.**

**The novelty of the design is the miniaturisation of the logger, which allows it to be used in smaller boreholes, and the change from a downward-facing PMT to a side-facing instrument. This first test shows promising results, which seem to require some adjustments to the sensitivity range and the borehole adapter.**

**The article should be published as it presents a test of a new instrument for in situ measurements in boreholes and its results. The description is well explained, but could go into more detail about the selected components and simulation/correction parameters.**

Reply: More details have been given in the ref. Schwander et al., 2023, but the RADIX system is too complex to provide a full component list in this brief communication.

**Specific comments:**

**From line 51 onwards, a correction factor of 0.017 is introduced for the "geometrical transmission", which was not taken into account in the previous work.**
**Could you specify all the parameters that are now taken into account?**

The main geometric parameters are given in the paper, i.e. the ratio of solid angles of the receiving PMT area and the LED radiation cone (0.017) and the estimated 5 dB lower signal due to different size and distance of receiver.

**For example, does this factor include the mentioned refraction? Does it vary with the type of drilling fluid and the angle of installation in the borehole adapter and/or the tilt in the borehole caused by the eccentrically mounted cable (shown in Figure 1)?**

There remains some uncertainty in the geometrical details (e.g. the focus of the mirror in front of the PMT and refraction of borehole wall), but since the main focus of the RADIX logger is to obtain a fast qualitative dust record of the deep part of an ice sheet for the purpose of age estimation and quality control of the stratification, the absolute signal level is not relevant.

**The logger signal shown in Figures 2 and 3 are plotted against borehole depth. Since Rongen (2020) has shown an optical anisotropy in ice, would switching to a directional optical measurement not require the addition of the orientation as a relevant dimension?**

**Rongen (2020) - Martin Rongen, Ryan Carlton Bay, and Summer Blot  The Cryosphere, 14, 2537–2543, 2020 https://doi.org/10.5194/tc-14-2537-2020 Observation of an optical anisotropy in the deep glacial ice at the geographic South Pole using a laser dust logger**

This is an interesting point. The anisotropy described by Rongen is from observations of light propagating over larger distances (125 m for IceCube and probably several meters in case of the laser dust logger) than in the case of our logger (approx. 0. 5 m). We expect therefore that our signal is much less affected by this anisotropy, but it would be certainly worthwhile to investigate. Furthermore, climatic dust level variations are fortunately larger and still allow to qualitatively assessing climate variations, in particular distinguishing warm interglacials (low dust content) from glacial conditions (high dust content)

**Authors responses to Reviewer3 Comments on egusphere-2024-372**

Reviewer 3: Peter Neff
* * *
"Brief communication: RADIX (Rapid Access Drilling and Ice eXtraction) dust logger test in the EastGRIP hole" by Jakob Schwander et al., The Cryosphere Discuss.
* * *
**This manuscript presents a test of a new RADIX borehole logging tool designed for the unique narrow diameter holes made by the RADIX device, using adapters to the larger diameter EGRIP borehole for testing. While I'm less equipped to comment on the specifics of the instrumentation and test, the manuscript clearly should be published after reviewing comments made by other reviewers. Some additional background on bubble ice to clathrate ice transition (and how its variable depth from site to site might affect logger performance) would be helpful for the reader, which could include references to the theoretical work of Miller (1969, Science), observations by Uchida and others which the authors are quite familiar with, or the ice core quality based review that I published in 2014 (though I don't insist at all on this self-reference!).**

Reply: This comment is appreciated and we have added Ref. Uchida et al. 2017, with some additional text.

**It will be very useful to see comparison (and validation of this test analysis) with the results from planned future testing in the Beyond EPICA borehole, but that obviously is beyond the scope of this manuscript. Hopefully there are RADIX boreholes to log as well in the near future.**

We fully agree with the reviewer.

---

## Editor Decision (ED1)

Schwander et al. 2024

Thank you for your revised submission, I am pleased to recommend publication subject to the following minor amendments, for improved understanding.

*Suggested amendments*

L14: Remove 'large' – this is subjective. If you want to note the size discrepancy between the borehole and the RADIX logger, I suggest 'In June 2023, we fitted the logger with an adapter to enable operation and testing in the deep EGRIP borehole'.

L15: switch 'excellent' for 'high quality' or 'error free' to reduce subjectivity

L26: join two sentences, to read 'It is designed to operate in bubble-free ice, since in bubbly ice the reflected light saturates the amplifier'.

L42: stray light – should be two words I think

L49: can you include an estimate of the concentration range? (even order of magnitude would be helpful for context)

L65: as above – order of magnitude estimate in brackets would be helpful

L96: comma after 'hoisting'

Figure 3: I would be interested in a brief sentence in the discussion on the discrepancy between RADIX and ABAKUS records at 1337.9m, but this is at your discretion.

---

## Author Response (AR2)

Thank you for your comments and suggestions, which we have taken into account in the new version.

Schwander et al. 2024

Thank you for your revised submission, I am pleased to recommend publication subject to the following minor amendments, for improved understanding.

*Suggested amendments*

*L14: Remove 'large' – this is subjective. If you want to note the size discrepancy between the borehole and the RADIX logger, I suggest 'In June 2023, we fitted the logger with an adapter to enable operation and testing in the deep EGRIP borehole'.*

done

*L15: switch 'excellent' for 'high quality' or 'error free' to reduce subjectivity*

Changed to 'high quality'

*L26: join two sentences, to read 'It is designed to operate in bubble-free ice, since in bubbly ice the reflected light saturates the amplifier'.*

done

*L42: stray light – should be two words I think*

done

*L49: can you include an estimate of the concentration range? (even order of magnitude would be helpful for context)*

(10 to 1000 ppb for the Antarctic plateau)

*L65: as above – order of magnitude estimate in brackets would be helpful*

 (100 to 10'000 ppb)

*L96: comma after 'hoisting'*

done

*Figure 3: I would be interested in a brief sentence in the discussion on the discrepancy between RADIX and ABAKUS records at 1337.9m, but this is at your discretion.*

The records shown in Fig. 3 exhibit several positive and negative discrepancies. Due to topographical surface features, we do not expect homogeneous dust deposition and thus homogeneous concentrations in horizontal layers. As we do not have precise information on these variations, we refrain from discussing the observed deviations.